# On Foundation Models for Dynamical Systems from Purely Synthetic Data

## Abstract

Foundation models have demonstrated remarkable generalization, data efficiency, and robustness properties across various domains. The success of these models is enabled by large-scale pretraining on Internet-scale datasets. These are available in fields like natural language processing and computer vision, but do not exist for dynamical systems. In this paper, we explore whether these properties can be achieved in the control domain when pretraining is performed entirely on synthetic data. We address this challenge by pretraining a transformer-based foundation model exclusively on synthetic data and propose to sample dynamics functions from a reproducing kernel Hilbert space. Our model, pretrained on purely synthetic data, generalizes to prediction tasks across different dynamical systems, which we validate in simulation and hardware experiments, including cart-pole and Furuta pendulum setups. Additionally, our model can be fine-tuned effectively to new systems increasing its performance even further. Our results demonstrate that even when pretrained solely on appropriately designed synthetic data, it is feasible to obtain foundation models for dynamical systems that outperform specialist models in terms of generalization, data efficiency, and robustness.

## 1 Introduction

Foundation models are trained on vast amounts of general data, allowing them to acquire extensive knowledge about a domain. This extensive pretraining enables them to easily adapt to new downstream tasks via no or little fine-tuning, often outperforming specialist models trained only for specific tasks (Bommasani et al., 2022; Brown et al., 2020). Specifically, foundation models have demonstrated unprecedented capabilities with regard to the following properties:

(**P1**) **Generalization**: The ability to perform well on a wide range of tasks, including those not explicitly trained on, by applying learned knowledge to new, unseen situations effectively (Yuan, 2023; Bommasani et al., 2022; Liu et al., 2024a).

(**P2**) **Data efficiency**: Using a pretrained foundation model on a new task requires small or no amounts of task-specific data, indicating the model's capacity to learn and adapt quickly without requiring extensive additional training data (Brown et al., 2020; Schick & Schütze, 2020; Liu et al., 2024a).

(**P3**) **Robustness**: Continuing to function reliably when faced with challenging conditions, such as distribution shifts, adversarial inputs, or noisy data, (Bommasani et al., 2022; Hendrycks et al., 2019).

Foundation models have been applied in various fields, where they are used to solve high-level tasks in settings such as vision (Kirillov et al., 2023; Liu et al., 2024b), language (Zhou et al., 2024), and robotics (Firoozi et al., 2024; Zeng et al., 2023). While robotics also deals with dynamical systems, existing foundation model research has focused on *multimodal* models and *high-level* tasks, which can be pretrained with Internet-scale text and image data, while leveraging general knowledge about the visual and semantic structure of the world (Brohan et al., 2022; Vuong et al., 2023; Shah et al., 2023). For instance, the RT-2 model integrates vision and language capabilities, enabling robots to interpret complex instructions and perform tasks like navigating to specific locations or manipulating objects based on visual cues. However, even the smallest version of RT-2, which has 5 billion parameters, can run at a frequency of around 5Hz (Zitkovich et al., 2023). This is

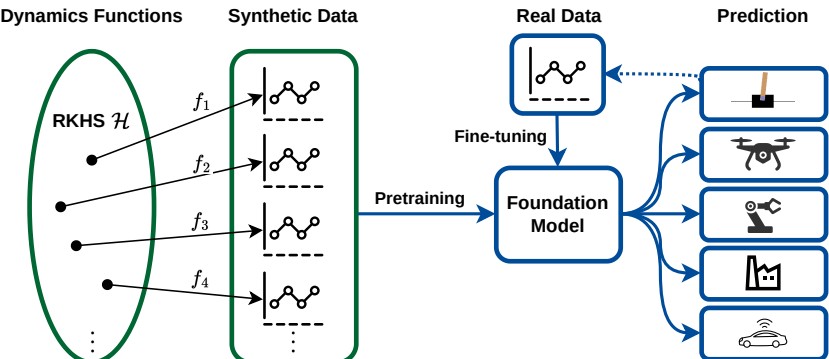

Figure 1: Proposed approach for a foundation model that predicts future states of dynamical systems. The model is pretrained purely on synthetic data (green). We propose to sample dynamics functions from an RKHS $\mathcal{H}$ and creating trajectory data by evolving the system through time. The foundation model is capable to zero-shot unseen systems in simulation and hardware (blue). Further, it can quickly be fine-tuned (dashed) to specific systems to improve the performance even further.

mainly because VLA models serialize images/text/actions into long token sequences (long contexts result in higher latency/memory). In the context of control, typical tasks such as state prediction and low-level feedback control require much faster update frequencies, and models that capture the inherent "dynamics" of the problem, much different from the knowledge represented in existing foundation models and available Internet-scale data. In this context, time-series transformers have the advantage of smaller contexts compared to VLAs, providing faster inference speeds.

In this work, we explore the feasibility of foundation models for dynamical systems that are tailored towards real-time, low-level control tasks. Further, the foundation models are trained entirely on carefully designed synthetic data. We consider the state prediction problem, i.e., the ability of a model to predict future states over a horizon, which underlies many control tasks such as feedback and state estimation. In this context, we show the typical foundation model benefits (**P1**)–(**P3**). In particular, we demonstrate that a foundation model pretrained solely on synthetic data can accurately predict the future states of a dynamical system different from those seen in pretraining, and that fine-tuning with little additional data can yield better performance and robustness than a specialist model trained on the specific task. Other foundation models build on the abundance of readily available datasets, e.g., from the Internet, (Radford et al., 2019; Brown et al., 2020). However, in control related dynamical systems, we lack such readily available Internet-scale data for training these models. Thus, we ask: *Can we train foundation models for control related dynamical systems from purely synthetic data and still realize their benefits?*

To address these challenges, we present a novel approach that leverages the TimesFM architecture (Das et al., 2024) and trains it from purely synthetic data representing dynamical systems. We propose to generate the data from dynamics functions sampled from a reproducing kernel Hilbert space (RKHS), well suited to capture a wide variety of dynamical behaviors (Ghojogh et al., 2021; Rosenfeld et al., 2024). Building on the resulting transformer-based foundation model, we investigate (and achieve) properties (**P1**)–(**P3**) for simulation and hardware examples. An overview of our proposed approach is illustrated in Figure 1. To summarize, our main contributions are:

- Pretraining on purely synthetic data based on RKHS dynamics functions;
- Validation of the foundation model claims (**P1**)–(**P3**) in simulation and on hardware; and
- Adaptation of the TimesFM architecture, including publishing our code and data.

By showing the feasibility of foundation models for dynamical systems, this work makes an important step for leveraging the power of this recent machine learning paradigm for control.

## 2    Related Work

To the best of our knowledge, there is only limited work on foundation models for dynamical systems and none of it addresses the question as we do. In the following, we describe the recent works Song et al. (2025); Seifner et al. (2024); Lai et al. (2025) that are closest to ours and have partially been developed in parallel. In contrast to Song et al. (2025), which requires known dynamics and initial solutions to accelerate simulations, our approach does not assume known dynamics. Similarly, while Seifner et al. (2024) focus on imputing missing time series data governed by ordinary differential equations (ODEs), our focus is forecasting multiple future states for control applications. In parallel to our work, Lai et al. (2025) introduce PANDA, a model trained on synthetic chaotic systems for scientific simulation tasks. In contrast, our approach targets a broader range of dynamical regimes beyond chaos, incorporates real-world evaluation on control-related systems, and explicitly investigates whether pretraining on synthetic data can generate the key properties of foundation models.

From a broader perspective, this work relates to three main areas: robotics, time series, and synthetic data pretraining. Similar to foundation models in robotics, we focus on real-world applications but aim to enhance real-time control capabilities beyond high-level task execution. This involves leveraging time series sensor data, which connects our research to time series foundation models. However, instead of focusing on repetitive trend forecasting, we concentrate on modeling system dynamics, typically exhibiting more complex behaviors. Additionally, we address the scarcity of large-scale real-world datasets by utilizing problem structure and expert knowledge to tailor the generation of synthetic data for pretraining. We elaborate on the three domains next.

In the domain of robotics, foundation models have predominantly concentrated on solving high-level tasks (Firoozi et al., 2024; Zeng et al., 2023). Specifically, their focus has been on leveraging text and image models to transfer general knowledge (such as object identification and understanding user text commands) to enhance robot perception. Consequently, current foundation models for robotics adopt multimodal approaches, incorporating additional data modalities to handle Vision-Language-Action scenarios (Vuong et al., 2023; Shah et al., 2023). Models like RT-2 exemplify this trend, processing image, text, and numerical data to facilitate exceptional perception and a more human-like understanding of the world (Brohan et al., 2022; Zitkovich et al., 2023). Similar architectures use specialized tokenization strategies, providing modularity to multiple data modalities but limiting inference speed (Schubert et al., 2023). However, the high dimensionality of the encoded input renders these models computationally intensive and slow in execution, making them unsuitable for low-level, high-speed, and real-time control tasks. In contrast, our work targets this limitation by developing foundation models from scratch for real-time control tasks, focusing on the ability to accurately model complex dynamical systems.

In the context of time series analysis, foundation models are predominantly applied to domains such as weather forecasting, Internet search trends, and energy usage, where the underlying patterns exhibit regular cycles and predictable behaviors. Notable examples include TimesFM (Das et al., 2024), Timer (Liu et al., 2024c), and TimeGPT (Garza & Mergenthaler-Canseco, 2023). Current work in time series emphasizes handling stable, cyclical data, often overlooking the complexities introduced by transient dynamics that characterize more volatile or rapidly changing systems. Additionally, most existing models focus on independent signals, disregarding the potential interactions and dependencies between multiple variables within a system (Nie et al., 2023; Zeng et al., 2022). This focus limits their applicability in scenarios where dynamics and interdependencies of the system variables are crucial. In contrast, our work diverges by prioritizing the modeling of intrinsic system dynamics, where time-dependent changes and interactions between variables are fundamental.

In both robotics and time series the pretraining of foundation models has largely relied on vast Internet-scale real-world datasets (Radford et al., 2019; Brown et al., 2020; Schneider et al., 2023; Bommasani et al., 2022). More recently, several studies have shown that synthetic data alone can serve as a powerful pretraining signal. In the time series domain, ForecastPFN, and TimePFN, based on TabPFN (Dooley et al., 2023; Taga et al., 2025; Hollmann et al., 2023), pretrain transformers entirely on synthetically generated temporal data that reproduce statistical properties such as trend and seasonality. These models demonstrate that large-scale synthetic pretraining can generalize across trend heavy forecasting tasks in both zero- and few-shot settings.

However, these synthetic datasets primarily emulate long-term statistical regularities of temporal data rather than the short-term dynamical behaviors characteristic of physical systems governed by underlying laws of motion. In contrast, our work introduces a form of synthetic pretraining grounded in system dynamics, generating data from RKHS dynamics functions. This represents a distinct synthetic data paradigm focused on capturing short-term, continuous-time evolution relevant to dynamical and control systems, rather than long-term temporal correlations.

Finally, recent works have also explored foundation models for solving partial differential equations (PDEs). Examples include multi-operator learning and extrapolation approaches (Sun et al., 2025), efficient foundation models for PDEs (Herde et al., 2024), and multimodal architectures for predicting multiple operators and symbolic expressions (Liu et al., 2024d). Generative and in-context pretraining strategies have also been proposed, such as ZEBRA (Serrano et al., 2024) and ICON (Yang et al., 2023; Yang & Osher, 2024). Parallel lines of work extend this idea beyond PDEs to symbolic and stochastic formulations of dynamical systems, where models aim to recover the underlying data-generating process from observed trajectories. Recent approaches infer symbolic ordinary differential equations (ODEs) from trajectory data (Becker et al., 2023; d'Ascoli et al., 2024), estimate drift and diffusion functions in stochastic differential equations (SDEs) (Seifner et al., 2025), or learn transition rates in Markov jump processes (MJPs) (Berghaus et al., 2024). In addition, He et al. (2025) investigate transformer-based architectures for large-scale chaotic system prediction, exploring potential of attention mechanisms for modeling complex dynamics. Despite their relevance in advancing operator learning and specialized architectures, these works address fundamentally different problem formulations - often focusing on specific PDE families or system identification in constrained dynamical settings - whereas our goal is to develop a general-purpose foundation model for real-time control across a broad spectrum of dynamical regimes, trained purely on synthetic data.

## 3 Problem Formulation

We aim to model the dynamics of discrete-time dynamical systems, defined by the transition function $f : \mathcal{X} \times \mathcal{U} \to \mathcal{X}$, where the state $x_{k+1}$ evolves as

$$x_{k+1} = f(x_k, u_k) + \varepsilon_k, \tag{1}$$

with process noise $\varepsilon_k$. The state space $\mathcal{X} \subseteq \mathbb{R}^{d_x}$ and action space $\mathcal{U} \subseteq \mathbb{R}^{d_u}$ represent the system's state and control input dimensions, respectively. We operate on finite-length trajectories consisting of state-action pairs, denoted $\mathbf{x} = \{(x_1, u_1), \ldots, (x_T, u_T)\}$. Subtrajectories are expressed using slicing notation, e.g., $\mathbf{x}_{k:k+H}$, representing the sequence from time step $k$ to $k + H$. This notation extends to sequences of only states or only actions, such as $x_{k:k+H}$ and $u_{k:k+H}$.

The goal is to train a foundation model $\mathcal{F}_\theta$ that generalizes across a family of dynamical systems that are randomly drawn from the space $\mathcal{H}$. For any $f \in \mathcal{H}$, the model should precisely predict future states based on past states and actions. Given a context window of length $c \in \mathbb{N}$ and a prediction horizon of length $m \in \mathbb{N}$, the model predicts the future states $\hat{x}_{k+1:k+m}$ using the context $\mathbf{x}_{k-c:k-1}$ and future actions $u_{k:k+m-1}$:

$$\hat{x}_{k+1:k+m} = \mathcal{F}_\theta(\mathbf{x}_{k-c:k-1}, u_{k:k+m-1}). \tag{2}$$

The objective is to minimize the prediction error for all $f \in \mathcal{H}$, ensuring robust and accurate performance across diverse dynamical systems. This state prediction problem essentially tests the capabilities of the model $\mathcal{F}_\theta$ to capture the relevant dynamics of a system, which is needed in basically all of model-based control. Explicitly, state prediction like (eq. (2)) is relevant in model-predictive control or state estimation, for example.

**Evaluation Objectives** We consider the evaluation objectives (**P1**)–(**P3**). We assess the model's ability to generalize by evaluating its mean squared error across a range of dynamical systems, including both seen and unseen systems from the family $\mathcal{H}$. The model's data efficiency is quantified by its performance across varying amounts of fine-tuning data. A data-efficient model achieves strong performance even when fine-tuned with minimal data or in zero-shot predictions (i.e., when the pretrained foundation model is used directly without any fine-tuning). Robustness is measured by the consistency of the model's performance across different

training runs and fine-tuning configurations. It is evaluated by analyzing the range of MSE values across multiple runs with different initializations and data subsets.

## 4 Synthetic Data Generation

The goal of our synthetic data generation is to obtain representative data that resemble general dynamical systems. In particular, we need a function space that is large enough, has easily adjustable properties such as smoothness, and can efficiently be sampled from. We propose to sample dynamics functions from an RKHS to create trajectory data. Kernel methods and RKHSs are frequently used in the context of dynamical systems (Pillonetto et al., 2014; Carè et al., 2023). Additionally, they have close connections to Gaussian process regression (Williams & Rasmussen, 2006; Kanagawa et al., 2018), which is often used to learn and represent the dynamics function (Buisson-Fenet et al., 2020; Treven et al., 2023; Svensson & Schön, 2017; Geist & Trimpe, 2020). Among other advantageous properties such as data efficiency and embedding of prior knowledge, uniform error bounds are tractable (Srinivas et al., 2012; Fiedler et al., 2021).

Whenever we represent dynamics functions with kernel methods, we obtain the approximation

$$f(x) = \sum_{i=1}^{n} \alpha_i k(x, x_i), \tag{3}$$

where $x_i$ are supporting (data) points and $\alpha_i$ coefficients obtained from the specific training method. There is a one-to-one correspondence between each RKHS $\mathcal{H}$ and the corresponding kernel function $k(.,.)$. Further, the norm of $f$ in the RKHS $\mathcal{H}$ is given by $\|f\|_{\mathcal{H}}^2 = \sum_{i=1}^{n} \sum_{j=1}^{n} \alpha_i \alpha_j k(x_i, x_j)$.

We use eq. (3) to sample functions from the RKHS. As a first design choice, we focus on dynamical systems without any control inputs, where the state dimension $d_x$ is four. The control is reintroduced later again in Sec. 6. Further, we break down the problem of sampling the vectorfield $f$ into sampling $d_x$ independent functions $f_i : \mathbb{R}^{d_x} \to \mathbb{R}$, where $f(x) = (f_1(x), \ldots, f_{d_x}(x))^\top$. We choose the standard RBF kernel $k(x, x') = \sigma^2 \exp(-\frac{\|x-x'\|^2}{2l^2})$ with hyperparameters $\sigma^2$ and $l$. We use RBF because it gives a single, tunable length scale and scales cleanly to large batches and in comparison with other alternatives such as the Matérn kernels, it provides smoother vector fields. We sample the supporting points $x_i$ uniformly from $[x_{\min}, x_{\max}]^{d_x}$ and coefficients $\alpha_i \sim \mathcal{N}(0, \sigma_\alpha^2)$ randomly, where $x_{\min}, x_{\max}, \sigma_\alpha^2$ are hyperparameters. To generate functions that represent a broad variety of behaviors, we scale the RKHS norm of each function to a target value selected from a uniform distribution between $\text{norm}_{\min}$ and $\text{norm}_{\max}$, enabling to sample functions in a specific norm range. We scale the coefficients $\alpha_i$ to ensure the function achieves the target norm $\alpha_i' = \alpha_i \cdot \frac{\text{norm}_{\text{target}}}{\|f\|_{\mathcal{H}}}$. Our approach of sampling RKHS functions is consistent with (Fiedler et al., 2021) and (Steinwart & Christmann, 2008).

We consider the sampled functions $f$ to correspond to a continuous time dynamical system. This way, we have more control over smoothness and discretization properties. To obtain trajectories, we use Euler's method to

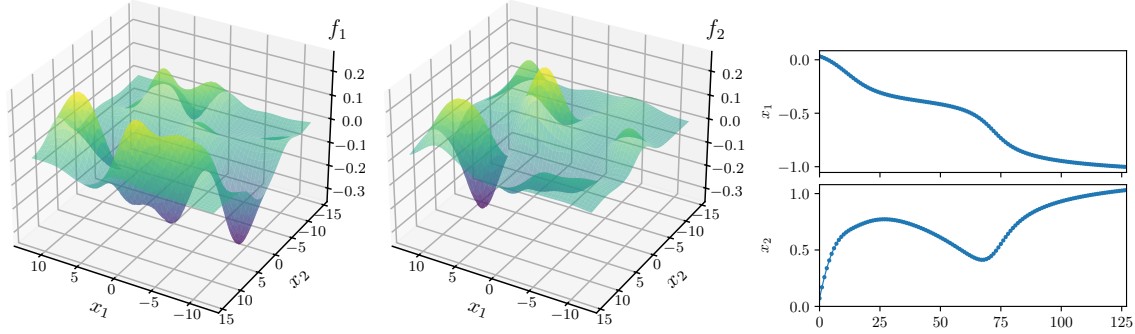

Figure 2: Representative functions sampled from an RKHS (left) in a two-dimensional state $(x_1, x_2)$ space and an example trajectory generated from the sampled dynamics functions (right).

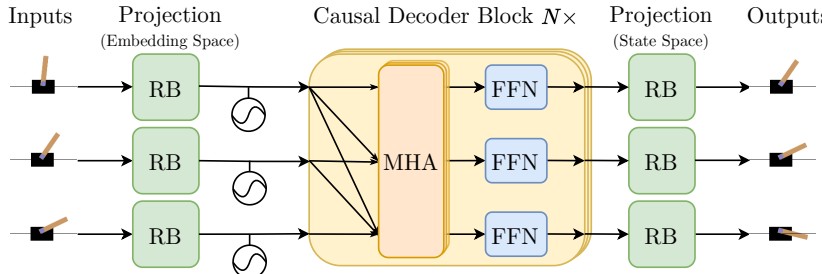

Figure 3: Architecture of the decoder-only transformer model. The model projects inputs into an embedding space using a residual block (RB), followed by $N$ causal decoder blocks with multi-head self-attention (MHA) and feed-forward networks (FFN). A final residual block projects from the embedding space to the output space.

discretize the system dynamics, where the next state is given by $x_{k+1} = x_k + \Delta t \cdot f(x_k)$. The initial state $x_0$ is sampled uniformly from $[0, 1]^{d_x}$, and the discretization step $\Delta t$ is chosen as a hyperparameter. In principle, any other method such as Runge-Kutta approaches could be used here as well.

Depending on the sampled dynamics and initial point $x_0$, data might look extremely similar. Thus, we propose a total variation selection criterion to remove overrepresented behavior and select trajectories, where the model can learn a meaningful general representation. The total variation of a trajectory $\mathbf{x} = \{x_0, x_1, \ldots, x_T\}$ is defined (Royden & Fitzpatrick, 2010) as $\mathrm{TV}(\mathbf{x}) = \sum_{k=1}^{T} \|x_k - x_{k-1}\|$. We select only those trajectories whose total variation $\mathrm{TV}(\mathbf{x})$ falls within the predefined bounds $\mathrm{TV}_{\min} \leq \mathrm{TV}(\mathbf{x}) \leq \mathrm{TV}_{\max}$. Finally, we bin the trajectories based on their total variation and downsample from overrepresented bins to ensure a balanced distribution of trajectories across the total variation spectrum. A representative function sample with its corresponding trajectory are shown in Figure 2.

We summarize our approach: (i) sample $N_s$ dynamics functions from the RKHS, (ii) scale each dynamics function's coefficients to a selected norm, (iii) sample a single trajectory from each dynamics functions using Euler's method, yielding $N_s$ trajectories, (iv) select trajectories based on a total variation threshold, and (v) bin trajectories based on total variation and downsample overrepresented bins. This procedure yields the final synthetic pretraining dataset consisting of $N$ trajectories. The corresponding data generation hyperparameters are provided in Appendix A.1.

## 5   Model

We utilize a decoder-only transformer model, as introduced by Das et al. (2024), which achieves state-of-the-art performance for time series foundation models. Specifically designed for time series forecasting, this architecture can effectively capture complex temporal dependencies and nonlinear dynamics, making it ideal for modeling dynamical systems.

**Model Architecture**   The model processes sequences of state-action pairs, $(x_k, u_k) \in \mathcal{X} \times \mathcal{U}$. Given an input trajectory $\{(x_1, u_1), \ldots, (x_T, u_T)\}$, it predicts a future trajectory $\{\hat{x}_2, \ldots, \hat{x}_{T+1}\}$. Only predictions $(\{\hat{x}_{c+1}, \ldots, \hat{x}_{c+m}\})$ beyond the observed context are used for evaluation. Formally, given

$$(\mathbf{x}_{\mathrm{input}}, \mathbf{u}_{\mathrm{input}}) = \{(x_1, u_1), \ldots, (x_c, u_c), (0, u_{c+1}), \ldots, (0, u_{c+m})\}, \tag{4}$$

where states beyond $c$ are set to zero as placeholders, the model generates $\{\hat{x}_2, \ldots, \hat{x}_{c+m+1}\}$, using only $\{\hat{x}_{c+1}, \ldots, \hat{x}_{c+m}\}$ as valid predictions for the true future states.

The overall model architecture is illustrated in Figure 3. Input state-action pairs $(x_i, u_i)$ are projected into a high-dimensional embedding space via a Residual Block (Das et al., 2023), expressed as $x_{i,\mathrm{embed}} = \mathrm{ResidualBlock}(x_i, u_i)$, capturing intricate system dynamics. The embedded sequence is then processed by a

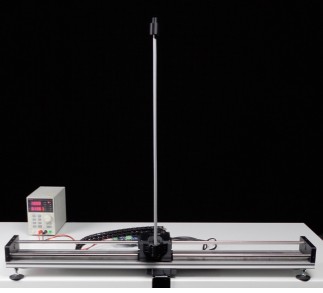 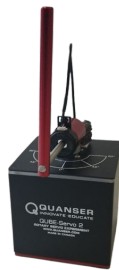

Figure 4: Custom cart-pole system on the left (image taken from (Anonymous, 2024)) and Quanser Qube Servo 2 Furuta systems on the right.

causal decoder with $N$ decoder blocks using self-attention (Vaswani et al., 2017). This causal design ensures predictions depend only on past and present inputs, enabling the capture of long-range dependencies to predict future states:

$$x_{0:c+m,\text{decoder}} = \text{decoder}(x_{0:c+m,\text{embed}}).  \tag{5}$$

**Model Training**   The model undergoes a two-phase training process: **pretraining** on synthetic data, as described in Section 4, to generalize across various dynamical systems, followed by optional **fine-tuning** on task-specific datasets using a reduced learning rate and fewer epochs for system-specific adaptation. We optimize the model using the AdamW optimizer (Loshchilov & Hutter, 2017), which combines adaptive learning rates with weight decay for stable and efficient training. Following Das et al. (2024), we increase the model's flexibility and robustness by employing random masking (masking 10% of input states) and output patching with a patch size of two while omitting input patching. This enables the model to predict two future states simultaneously during training, and focus on the actual dynamics of the series.

**Data Augmentation**   We apply data augmentation techniques to improve model training and generalization, distinguishing between synthetic data during pretraining and dynamical systems data during fine-tuning. **During pretraining**, we apply scaling and shifting (Wen et al., 2020) to trajectories: $\mathbf{x}' = \alpha \mathbf{x} + \beta$, where $\alpha \in [\alpha_{\min}, \alpha_{\max}]$ and $\beta \in [\beta_{\min}, \beta_{\max}]$ are randomly sampled from uniform distributions. **During fine-tuning**, we add Gaussian noise (iid) to each state measurement of the dynamical systems data: $x'_k = x_k + \varepsilon_k$, where $\varepsilon_k \sim \mathcal{N}(0, \sigma^2)$.

# 6   Experiments

In this section, we will validate that pretraining our transformer model on RKHS data yields the typical foundation model properties (**P1**)–(**P3**).[1] In particular, we provide evidence that: (i) our model generalizes, i.e., it is able to accurately predict future states of completely unseen systems, indicating that the transformer architecture is a good choice overall; (ii) learning is more robust, i.e., our synthetically pretrained transformer model performs best and additionally shows significantly lower variance than all other baselines; and (iii) we are more data efficient, i.e., the loss of the synthetically pretrained model is substantially lower than of all other models when new data is used for fine-tuning instead of learning from scratch.

We evaluate the proposed pipeline on both simulation and hardware systems. Our transformer model, comprising 20 layers and approximately 3.4 million parameters (see Appendix B), is pretrained on synthetic RKHS data and optionally fine-tuned on dynamical systems data. This yields two models: the **pretrained model (Pre)** and the **fine-tuned model (Ft)**, which are evaluated on test sets and compared against baselines. Each experiment is repeated 20 times for statistical significance. All scratch-trained (LT) and fine-tuned (Ft) models use the same fine-tuning schedule (optimizer, batch size, epochs/steps, and LR schedule). Thus, any differences reflect initialization (pretraining) rather than longer training.

---

[1]The code and data from this work can be downloaded from Anonymous Repository.

**Datasets** We use both simulated and real-world datasets for training and evaluation. The *simulated datasets* are: (i) a cart-pole simulation with fixed parameters and constant action of zero, where the pole is randomly initialized near the upright position and swings freely, and (ii) a cart-pole simulation with randomized parameters and pink noise action, with each trajectory reflecting a unique system configuration. The *hardware datasets* are: (iii) a custom cart-pole system with action, whose parameters are described in (Anonymous, 2024), and (iv) a Furuta pendulum using the Quanser Qube Servo 2 (Quanser, 2024). Both hardware systems are shown in Figure 4. Each trajectory is $d = 4$-dimensional, with context length 32 and a prediction horizon 32, reflecting our focus on short-horizon prediction for control rather than long-range forecasting. Unless stated otherwise, train/val/test splits are 80/10/10 with 20,000 training trajectories per dataset. More details on the datasets used for evaluation can be found in Appendix A.

## 6.1 Experimental Setup

**Evaluation Metrics** We evaluate our models based on the defined objectives (**P1**)–(**P3**). The evaluation metric for generalization is the mean squared error (MSE) over the trajectory. For the unseen part of the trajectory (i.e., the prediction horizon) of length $m$, we calculate the MSE between the predicted states $\hat{\mathbf{x}}_t$ and true states $\mathbf{x}_t$ as MSE $= \frac{1}{m} \sum_{t=1}^{m} \|\hat{\mathbf{x}}_t - \mathbf{x}_t\|^2$. Furthermore, we assess the robustness of the model by evaluating the range of MSE values across the worst and best runs. A smaller range indicates a more stable and robust model. To investigate data efficiency, we conduct experiments using datasets of varying sizes: 2%, 10%, and 100% of the total 20,000 trajectories for each training dataset and compare the MSE across these data levels. In this setup, the pretrained model is treated as having seen 0% of the data, as it has not seen any data from the cart-pole system. We compare the performance of our proposed model against several baselines:

- **Linear Regression (LR)**: A classical method with limitations in capturing nonlinearity.
- **Feedforward Neural Network (FNN)**: A small neural network with three layers consisting of 128, 64, and 32 neurons, respectively, using ReLU activation functions.
- **Smaller Transformer (ST)**: A reduced version of the proposed transformer model (without pretraining), using the same decoder-only architecture but with 8 layers and approximately 200k parameters.
- **Large Transformer (LT)**: The proposed transformer model (without pretraining), of 20 layers and approximately 3.4 million parameters. The LT baseline is not pretrained, serving as an ablation of the pretraining and thus shows the impact of it and the foundation model paradigm.

We use LR and FNN as sanity check references. Since they lack the causality mechanism inherent in transformer architectures, we apply an iterative prediction approach. In this setup, the predicted state is fed back into the model to predict the subsequent state, resulting in the prediction formulation $\mathbf{x}_{k:k+31} \rightsquigarrow \hat{x}_{k+32}^{\mathrm{LR}}$, where the model takes state-action pairs from steps $k$ to $k+31$ as input to predict the state at step $k+32$.

In addition, we compare our approach with the zero-shot performance of **TimesFM** (Das et al., 2024) and **Timer** (Liu et al., 2024c), two state-of-the-art foundation models for time-series forecasting pretrained on large-scale trend-heavy datasets. For control-oriented applications, however, the relevant dynamics typically unfold over short horizons, where a limited temporal context is sufficient to capture the underlying behavior. Consequently, we consider a setting in which the models predict 32 time steps given 32 steps of context. In contrast, standard time-series foundation models are designed for long-term forecasting and rely on substantially larger patch lengths - 96 for **Timer** and 32 for **TimesFM** - to capture extended trends, although they are trained to also allow the usage of prepadded sequences. To ensure a fair comparison in terms of available data, we provide each model with the same effective sequence length, zero-padding shorter contexts as needed to match their respective patch lengths. This comparison emphasizes the core idea of our work: that pretraining on synthetically generated trajectories from a rich RKHS sampling of dynamical functions (capturing diverse behaviors and regimes beyond the trends seen in typical large-scale datasets) can yield models that exhibiting the foundation model properties (**P1**)–(**P3**) in a dynamical systems setting. Further, it is not sufficient to train on trend-heavy datasets.

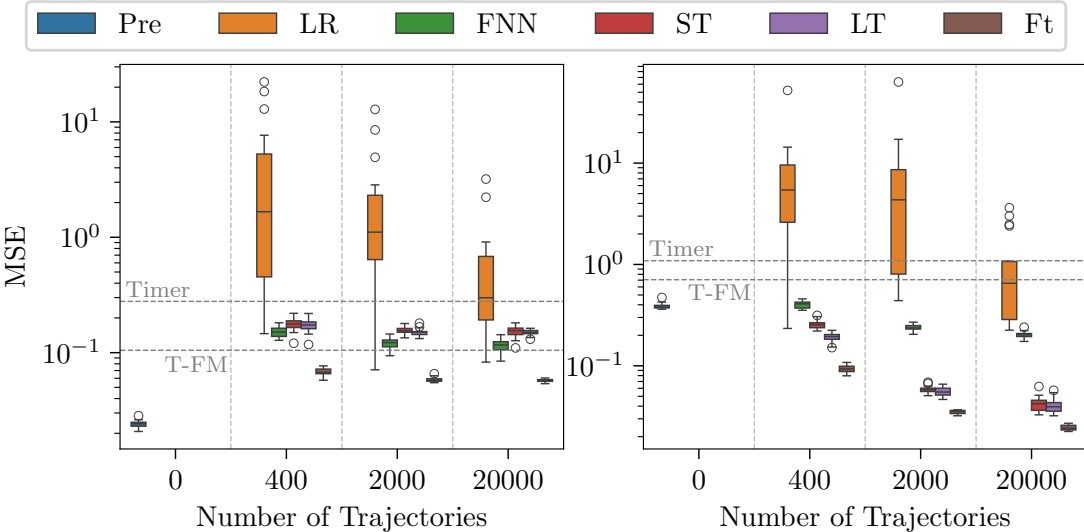

Figure 5: Simulation results for the cart-pole system. Left: fixed parameters with constant action. Right: randomized parameters with pink noise action. The plots show the MSE of our models (Pre, Ft) outperform the baselines (LR, FNN, ST, LT). Similarly, the Timer (Liu et al., 2024c) and TimesFM (T-FM (Das et al., 2024) baselines are shown as dotted lines. in both settings being outperformed in zero-short settings by our synthetic data–pretrained models.

## 6.2 Simulation Cart-Pole Experiments

Figure 5 shows the simulation results. Across both experiments, the transformer models exhibit good generalization, as indicated by low MSE values. Notably, the pretrained and fine-tuned models outperform the baselines—including both the smaller and larger transformer variants—demonstrating the advantages of pretraining on synthetic data. Comparing LT (scratch) and Ft (pretrained and fine-tuned) under a matched fine-tuning budget, Ft attains lower test MSE and smaller variation in the 2% and 10% data regimes; the gap narrows at 100%, consistent with diminishing returns from pretraining as more target data becomes available.

Furthermore, the zero-shot pretrained transformer model (Pre) achieves reasonable performance despite not having seen any cart-pole data. In the fixed parameter experiment, it outperforms the linear regression baseline and performs comparably to the feedforward neural network in low-data scenarios, highlighting its effectiveness in zero-shot prediction settings. In the randomized parameters experiment with pink noise, although the pretrained model's performance is reduced (likely due to random actions introducing unseen dynamics), it still provides a robust initialization that contributes to the superior performance and robustness of the fine-tuned model.

In addition, the zero-shot performance of our model, pretrained on purely synthetic data, outperforms the zero-shot performance of both **TimesFM** (Das et al., 2024) and **Timer** (Liu et al., 2024c), shown as dotted lines in Figure 5. Despite being pretrained on large-scale, trend-heavy Internet datasets, both models are consistently outperformed by our synthetic data–pretrained transformer, especially in the randomized parameters setting with pink noise ac-

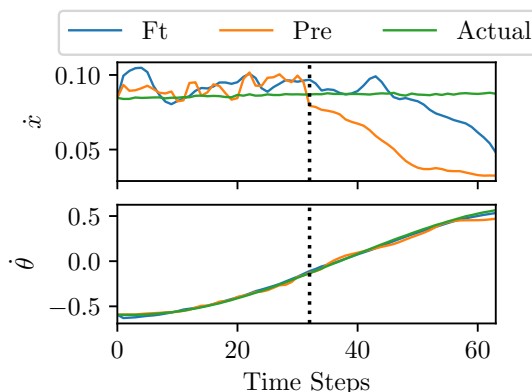

Figure 6: Example trajectory for the cart-pole system with fixed parameters and constant action. The plot compares the pretrained and fine-tuned models with the actual $\dot{x}$ and $\dot{\theta}$.

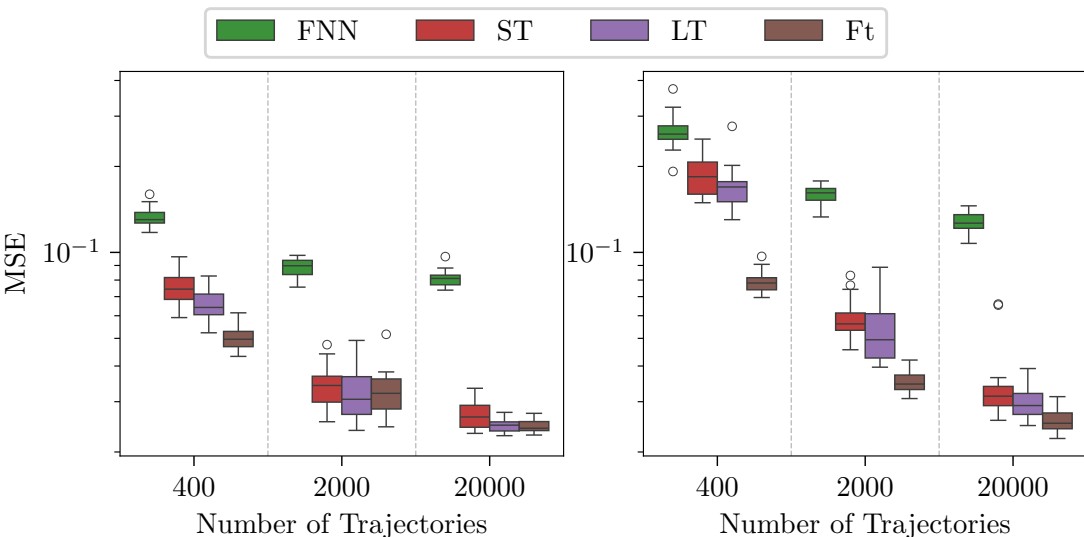

Figure 7: Hardware results for the cart-pole (left) and Furuta pendulum (right). The plots compare the MSE of the fine-tuned model (Ft) against baselines (FNN, ST, LT) across data subsets, highlighting generalization, data efficiency, and robustness of our approach.

tions. This suggests that pretraining on synthetic trajectories tailored to dynamical systems can provide more relevant inductive biases than large-scale generic time-series data, particularly when system dynamics are driven by non-constant control inputs. This advantage is of special importance, as the acquisition of relevant real world data is expensive in comparison with the generation of well chosen synthetic data.

In Figure 5, at low data (2%, 10%), **Pre** can match/exceed **Ft**. This can be due to *feature distortion*: narrow, fine-tuning under distribution shift can erode robust features learned at pretraining making the model more susceptible to OOD or any mismatch between training and testing data; the effect diminishes as training data grows (Kumar et al., 2022; Wortsman et al., 2022).

Figure 6 presents a specific cart-pole trajectory with fixed parameters and constant action, comparing the pretrained and fine-tuned models against the actual velocities $\dot{x}$ and angular velocity $\dot{\theta}$ (note the different y-axis scalings). This trajectory demonstrates the strength of the pretrained model and the improvement after fine-tuning.

### 6.3 Hardware Cart-Pole and Furuta Experiments

Results from applying the foundation model on actual experimental data are shown in Figure 7. We exclude linear regression due to its poor performance, and zero-shot performance (Pre) was omitted since pretraining considered zero action ($u \equiv 0$), which is not the case here. Action is highly relevant in these experiments, and Ft can adapt to it.

For the hardware cart-pole with the full dataset (20,000 trajectories), the best run of the large transformer (LT) slightly outperforms the fine-tuned transformer (Ft), indicating that a large amount of data can partially offset the benefits of pretraining. However, the median performance of the fine-tuned transformer remains superior, highlighting its reliability and data efficiency. Additionally, the fine-tuned transformer shows robust performance, as evidenced by its narrow range of MSE values across runs.

The results for the hardware Furuta pendulum system follow similar trends. While all baseline models have a high range of MSE values across runs, the fine-tuned transformer (Ft) consistently show a narrow range, reflecting its stable and reliable performance. This robustness is maintained across all data subsets, suggesting that increasing data alone does not resolve the variability observed in the baselines.

Figure 8 presents a specific trajectory of the hardware Furuta system. The plot compares the fine-tuned model with the FNN baseline and the actual trajectory, showcasing that the fine-tuned model closely follows the real dynamics and outperforms the FNN baseline.

Overall, the experimental results support the fact that the foundation model pretrained with synthetic data shows: **(P1) generalization**, which is demonstrated by the consistently low MSE observed in both simulation (fixed and randomized experiments) and hardware tests, indicating that our model accurately predicts unseen dynamics within the studied scope; **(P2) data efficiency**, which relates to the competitive performance achieved even with limited training data, as seen in the zero-shot and low-data scenarios; and **(P3) robust performance**, which is reflected in the low variance of MSE across different runs and setups, confirming the stability of our approach in both simulated and real-world conditions.

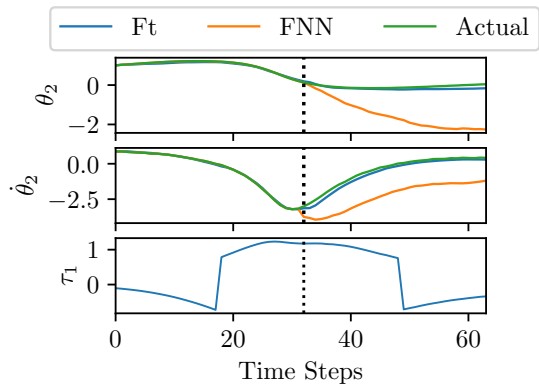

Figure 8: Hardware trajectory for the Furuta pendulum system. The plot compares the fine-tuned model (Ft) against the FNN baseline and the arm angle $\theta_2$ and angular velocity $\dot{\theta}_2$ (with control action $\tau_1$).

## 7 Conclusion

This work investigated the viability of foundation models for state predictions in dynamical systems using a transformer-based approach entirely trained on carefully designed synthetic data. In this context, we show that foundation models achieve: **(P1)** effective generalization to unseen systems, **(P2)** data efficiency enabled by pretraining on synthetic data, and **(P3)** robust performance across varying conditions. In our simulation experiments, the synthetic data pretrained and finetuned model consistently outperforms baseline methods, achieving lower MSE and reduced performance variance. Hardware experiments further supported these findings by showing the model's ability to adapt to real-world control tasks with minimal fine-tuning. The observed zero-shot performance and subsequent improvements after fine-tuning highlight the role of synthetic data pretraining in enhancing **(P1)**–**(P3)**. These results contribute toward future research in real-time control systems and the development of generalist models for a broad range of dynamical applications. In summary, our work provides a compelling demonstration that *synthetic data* pretraining can serve as a robust foundation for scalable and adaptive models in control applications.

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

# A   Datasets

## A.1   RKHS Dataset

Table 1: Hyperparameters for RKHS-based synthetic dynamics dataset.

| Parameter | Symbol | Value |
|---|---|---|
| Number of trajectories | $N$ | 99,920 |
| State dimension | $d_x$ | 4 |
| RBF kernel variance | $\sigma^2$ | 4.0 |
| RBF kernel length scale | $l$ | 1.5 |
| Number of basis functions | $n$ | 200 |
| Coefficient variance | $\sigma_\alpha^2$ | 1.0 |
| Supporting point bounds | $x_{\min}$ | -10.0 |
| | $x_{\max}$ | 10.0 |
| RKHS norm range | $\text{norm}_{\min}$ | 5.0 |
| | $\text{norm}_{\max}$ | 10.0 |
| Total variation bounds | $\text{TV}_{\min}$ | 0.5 |
| | $\text{TV}_{\max}$ | 3.75 |

## A.2   Cartpole Simulation (fixed parameters)

Table 2: Simulation parameters and dataset split for fixed-parameter cartpole.

| Parameter | Symbol | Value |
|---|---|---|
| Number of trajectories | $N$ | 25,000 |
| | $N_{\text{train}}$ | * |
| | $N_{\text{val}}$ | 2,500 |
| | $N_{\text{test}}$ | 2,500 |
| Gravity constant | $g$ | $9.8\,\text{m/s}^2$ |
| Mass of cart | $m_{\text{cart}}$ | $1.0\,\text{kg}$ |
| Mass of pole | $m_{\text{pole}}$ | $0.1\,\text{kg}$ |
| Length of pole | $l_{\text{pole}}$ | $1.0\,\text{m}$ |

## A.3   Cartpole Simulation (random parameters)

Table 3: Parameter distributions and dataset split for random-parameter cartpole.

| Parameter | Symbol | Range / Value |
|---|---|---|
| Number of trajectories | $N$ | 25,000 |
| | $N_{\text{train}}$ | * |
| | $N_{\text{val}}$ | 2,500 |
| | $N_{\text{test}}$ | 2,500 |
| Gravity constant | $g$ | $9.8\,\text{m/s}^2$ |
| Mass of cart | $m_{\text{cart}}$ | $\mathcal{U}(0.7,\,1.3)\,\text{kg}$ |
| Mass of pole | $m_{\text{pole}}$ | $\mathcal{U}(0.05,\,0.15)\,\text{kg}$ |
| Length of pole | $l_{\text{pole}}$ | $\mathcal{U}(0.3,\,0.7)\,\text{m}$ |
| Maximum force magnitude | $|F|_{\max}$ | $\mathcal{U}(8.0,\,12.0)\,\text{N}$ |

### A.4 Cartpole (real)

Table 4: Dataset split and statistics for real-world cartpole experiments.

| Parameter | Symbol | Value |
|---|---|---|
| Number of trajectories | $N$ | 25,000 |
| | $N_{\text{train}}$ | * |
| | $N_{\text{val}}$ | 2,500 |
| | $N_{\text{test}}$ | 2,500 |

### A.5 Furuta (real)

Table 5: Dataset split and statistics for real-world Furuta pendulum experiments.

| Parameter | Symbol | Value |
|---|---|---|
| Number of trajectories | $N$ | 25,000 |
| | $N_{\text{train}}$ | * |
| | $N_{\text{val}}$ | 2,500 |
| | $N_{\text{test}}$ | 2,500 |

## B  Model Architecture

Table 6: Transformer architecture hyperparameters.

| Parameter | Value |
|---|---|
| Residual hidden dimension | 256 |
| Embedding dimension | 64 |
| Number of attention heads | 64 |
| Number of transformer layers | 20 |
| Transformer hidden dimension | 1024 |

