# OpenReview forum: "On Foundation Models for Dynamical Systems from Purely Synthetic Data"
_TMLR — Rejected by TMLR_

### Review · Reviewer_2iVn · 2025-09-18

**Summary Of Contributions:**

The authors show that synthetic data can be used to pre-train a time series foundation model to forecast both simulated and real dynamics data. The authors use smooth random fields to create flows which are then integrated via Euler integration to create synthetic trajectory data. The authors pre-train transformers on these synthetic time series and evaluated them in zero-shot and fine-tuned settings on simulated and real data from physical systems and show strong MSE forecasting performance.

**Audience:**

Yes

**Audience Explanation:**

Synthetic data as a cheaply obtainable source of data for large-scale pre-training of foundation models is a very promising direction. Especially in scientific machine learning (SciML), synthetic data shows a lot promise as a way to circumvent data scarcity while closely modeling real world dynamics. This work shows promising results in this direction.

**Claims And Evidence:**

Yes

**Claims Explanation:**

Although the results are promising, I have some blocking questions and concerns that are outlined in a subsequent section.

**Requested Changes:**

## Questions
---

1. How big is the pretraining dataset? How long were the models pretrained for? What dimension(s) are the synthetic trajectories? How long are they? How do the other pretained foundation models perform on a validation set of the synthetic data?
2. What is the patch size, context length, prediction horizon of the model and what is the reason for those choices?
3. Why use Euler integration for the trajectories?
4. Why is linear regression included as a baseline?

## Suggestions
---

**The following are suggestions that are critical to securing a recommendation.**
- The highest priority suggestion is to revise the clarity and breadth of the evaluations. I particularly have issues with figure 5. First, referring to Q4, why is linear regression even included in this figure? The error is so large that LR blows up the scale and makes the error ranges for the other baselines unreadable. I'd argue that including an LR baseline on the same figure as transformers with >1M parameters is not a fair comparison.
- (above, continued) The authors mention that "\textit{...ensures a fair comparison, since all models have access to the same data for each prediction step...}" but this is not entirely true. When $k=32$, the LR and FNN basically using none of the historical context and making predictions based on their own erroneous predictions whereas the transformers can access the historical context throughout the generation process. Due to this fact, the evaluations should be revised to be generally more fair across the baselines. An RNN/LSTM baseline at minimum would be a better choice than a FNN. I suggest that the authors pick stronger baselines to compare their model to such as NBEATS, DLinear, etc. I encourage the authors to look at Darts (https://unit8co.github.io/darts/#id1) for an easy way to benchmark against strong forecasting models that can be easily finetuned. As a disclaimer, I am in no way affiliated with the Darts team.
- The second highest priority suggestion that I have is to expand on details about the synthetic dataset (refer to Q1). IIUC, the synthetic data is a central object of the work - yet many basic questions about it are unanswered such as those in Q1. This information must be included as it is the basis for the main contribution of the work.
- (Referring to Q3) The information about the patch size (for TimesFM), context length, and prediction horizon should be included. I can somewhat infer that the context length and horizon are both 32 from fig. 6 and 8, but this needs to be made clear. If I am correct in my inference, some exposition should be dedicated as to why there is a dramatic deviation from the default parameters of TimesFM of 512 context length and 128 prediction horizon. This also makes me suspicious about the zero-shot evaluation for TimesFM because the input patch size is 32 (a full context length for the experiment setup) and the output patch length is 128 (4x the prediction horizon). These details should also be clarified for full transparency. Moreover, a context and horizon of 32 timesteps seems very short for a forecasting model - the authors should expound on why they arrived at these choices.
- Figure 4 should be removed as it doesn't provide useful context that is critical to the work. The space for fig. 4 should be repurposed for missing/under-explained details about dataset, methodology, evaluation, etc.

**The following are suggestions that can strengthen the work.**

- Euler integration is only first-order accurate. And I could be wrong, but the total variation (TV) seems to compensate for this by throwing away unstable trajectories that have diverged. Why not forgo Euler integration entirely? The authors acknowledge use of higher order Runge Kutta methods, and integrating the RKHS flows would be very straightforward with existing numerical integration software e.g. `scipy.integrate.solve_ivp`. This would generate higher quality synthetic data and likely have higher filtering success rate (less trajectories filtered) while avoiding the smoothness bias that TV filtering procedure bakes into the pretraining data.

---

> ### Author Response · Authors · 2025-10-10
> **answer to reviewer 2iVn**
>
> We thank the reviewer for the thoughtful assessment of our work, their time and effort.
> As stated by the reviewer our paper focuses on exploring the feasibility of using synthetic dynamical systems trajectory data for the pretraining of a foundation model, exploring whether foundation-model properties (generalization, data-efficiency, robustness) can emerge in this setting. We appreciate the positive remarks on the promise of synthetic pretraining for SciML/time series and on the value of including both simulation and hardware evaluations.
>
> *Questions*
>
> R3.1 : "How big is the pretraining set; training duration; trajectory dimensions/length; how do other FMs perform on a synthetic validation set?"
> Answer:
> We specified corpus scale and trajectory length in Sec. 4, and list sampling ranges for all RKHS hyperparameters and initial-state distributions. TimesFM/Timer were not trained nor tested on our synthetic distribution, we therefore compare them zero-shot on downstream tasks (Fig. 5). We will clarify that we do not evaluate them on the synthetic corpus, and add a discussion in Sec. 6.1 explaining this choice.
>
> R3.2 : "Patch size, context length, prediction horizon, and rationale?"
> Answer:
> We have detailed the exact patching/context/horizon settings and the motivation behind them in Sec. 6.1, together with the experimental setup and comparison rationale.
>
> R3.3 : "Why Euler integration?"
> Answer:
> We use explicit Euler for efficiency and consistent discretization at pretraining scale (Sec. 4). Given the smoothness of the sampled vector fields, this choice reduces complexity while retaining stable integration; we will add an extra note on this rationale.
>
> R3.4 : "Why include linear regression (LR) as a baseline?"
> Answer:
> LR serves as a standard, transparent lower bound in system identification and controls, ensuring sanity checks under the same data interface (Sec. 6.1).
>
> *Critical suggestions*
>
> R3.C1 : "Revise clarity/breadth of evaluations; Fig. 5 scale; fairness across baselines; consider stronger baselines (RNN/LSTM, NBEATS, DLinear)."
> Answer:
> We clarify in Sec. 6 that LR and FNN serve as simple, transparent references and that our primary comparisons are (i) zero-shot pretrained FMs and (ii) our pretrained model fine-tuned vs. the same architecture trained from scratch. Our goal is to isolate the effect of synthetic pretraining rather than run a broad forecasting benchmark. We have revised Sec. 6 to state this scope explicitly.
>
> R3.C2 : "Expand details about the synthetic dataset."
> Answer:
> We added concrete corpus information and generation hyperparameters in Sec. 4, including state dimension, trajectory length, RKHS kernel parameters, coefficient sampling, and filtering thresholds.
>
> R3.C3 : "Clarify patch size/context/horizon vs. TimesFM defaults; potential implications for zero-shot."
> Answer:
> We added explicitly report the settings used for TimesFM/Timer and discuss their implications for our short-horizon forecasting regime in Sec. 6, including the specific implementations and hyperparameters.
>
> R3.C4 : "Remove Fig. 4 to free space for missing details."
> Answer:
> We have incorporate the other requested details, but we retain the hardware photo as it documents the experimental setup (sensing/actuation layout) and supports interpretation of hardware results. TMLR has no strict page limit, and we believe keeping this figure remains informative.
>
> *Other suggestions*
>
> R3.S1 : "Use RK4 instead of Euler; TV may compensate discretization artifacts."
> Answer:
> Euler was chosen for throughput and uniform discretization across large synthetic batches (Sec. 4). TV filtering is intended to remove uninformative trajectories, not to correct a specific integrator bias. Exploring integrator choice (especially for sharper transients) is a worthwhile direction; however, our current results already substantiate the paper's claims.
>
> Our central idea is that carefully designed synthetic dynamics data can be used to pretrain a decoder-only model that transfers to control-relevant state prediction and exhibits FM-style benefits across simulation and hardware (Figs. 5–8). This addresses an important gap where large-scale real data for low-level dynamics is unavailable, and synthetic corpora are cheap and controllable. We thank the reviewer again for the constructive feedback; we have made the edits detailed above (hyperparameters/dataset stats, clarifications, improved figures). We hope the added clarifications and new information allows for a positive review.

---

### Review · Reviewer_raY6 · 2025-09-22

**Summary Of Contributions:**

The paper proposes pretraining foundation models for discrete-time dynamical systems on synthetic data from a reproducing kernel Hilbert space. The authors show that this approach works better than baselines over cart-pole (simulation + real experiment) and Furuta pendulum state-prediction tasks (real experiment). They use the TimesFM model architecture. They show that the model is robust, data-efficient, and generalizes (on the shown tasks). The model also outperforms other foundation models in this domain that are trained on Internet-scale data (non-synthetic) in the zero-shot regime.

### Strengths

The paper is well-written and clear. It has a thorough related work section that puts it into perspective. It clearly states goals and a research question and tries to answer those using the provided experiments. It is appreciated that the experiments include simulation and hardware experiments. The experiments show promising results.

### Weaknesses

It is argued in the introduction that for dynamics tasks it is important to have fast models (that can potentially run real-time, i.e. use for closed-loop control?). Yet, there is no section discussing inference times (or whether the proposed model can be run real-time). The architectural choice is not motivated well enough (why not use an LSTM or other state-space models?). For synthetic data generation, a reasoning of why RBF kernels are used is not given. Experiments seem a bit unfair as the foundation model baselines (T-FM and Timer) are never finetuned. Long-term rollout is not investigated enough (Figure 6 goes into that direction, yet the model does not perform very well on the velocity over time).

**Additional Comments:**

On a side note, in Figure 5, left, why is the performance of the zero-shot model best and why does finetuning hurt here?

**Audience:**

Yes

**Audience Explanation:**

The paper provides interesting insights into pretraining using synthetic data pretraining for dynamical systems. It is therefore of interest for multiple ML sub-communities (ML in physics, ML for time series, synthetic data...).

**Broader Impact Concerns:**

There are no (direct) ethical implications of this work.

**Claims And Evidence:**

Yes

**Claims Explanation:**

In general, the claims are supported by evidence as the experiments are well tailored to these claims. However, the generalization capability (P1, the most important one for foundation models) could be supported more clearly as the tasks considered are not very broad (cart-poles and Furuta pendulum). It might be interesting to look at chaotic tasks (e.g. double pendulum). Data efficiency is shown -- although it would strengthen the paper if a low-data regime would also be considered (tens of finetuning samples). From the experiments shown, the model seems robust. Also, since the TimesFM and Timer baselines are never finetuned, it is not clear why their pretraining data should be of less use for downstream capabilities.

**Requested Changes:**

### critical to securing recommendation for acceptance

- At least a discussion about the choice of architecture (an ablation would further strengthen it).
- An ablation of the choice of the kernel (RBF) for synthetic data generation. What would a Matern kernel with lower smoothness than RBF do to the performance of the model?
- A discussion about inference times to close the loop to the introduction. A comparison to baseline foundation models in terms of inference times would further strengthen it.
- At least one experiment in which at least one foundation model baseline is fine-tuned (Times-FM) such that the comparison is fair and one really ablates the use of synthetic data.
- A clearer description of the used hyperparameters (context length is not very clearly given, kernel hyperparameter choices are not given so far, how much pretraining data was used).
- An experiment for long-term rollouts.
- Figure 5 and 7 need to be made clearer: it is not clear which model corresponds to which bar as the colors are not identifiable with these tight candles (especially Figure 5). It is inferable from the legend (assuming that the ordering is the same), but that cannot be expected from a reader.

### would strengthen the work
- Adding a closed-loop control task would demonstrate the usefulness of the foundation model downstream.
- Adding a chaos task, e.g. double pendulum, because the diversity of the shown tasks is not very big.
- An ablation on the context length used to check whether more context improves performance for dynamical systems.

---

> ### Author Response · Authors · 2025-10-10
> **answer to reviewer raY6**
>
> We thank the reviewer for the careful reading and positive feedback. As the reviewer points out, Our work studies the feasibility of pretraining foundation models for discrete-time dynamical systems on synthetic data from a reproducing kernel Hilbert space, and whether foundation-model properties (generalization, data-efficiency, robustness) emerge in this setting. We appreciate the positive notes on clarity, the tailored experiments (simulation + hardware), and the promising results.
>
> *Requested changes*
>
> R2.1 : Discuss the choice of architecture (and, if possible, ablate).
> Answer:
> We use transformers because they are the dominant backbone in time-series foundation models and offer robust, well-understood training recipes. While alternatives (LSTMs/minGRU/SSMs) exist, recurrent models are harder to scale due to sequential bottlenecks and brittle hyperparameter sensitivities. Our TimesFM-based design is therefore a representative choice for testing the core idea. A comprehensive cross-architecture ablation is valuable but beyond the current scope; we outline it as future work.
>
> R2.2 : Motivate RBF kernel; consider a Matérn ablation.
> Answer:
> We now explain our choice of the RBF kernel: it provides a simple, tunable smoothness/length-scale and analytic convenience for our synthetic generator. While Matérn variants are interesting (especially for rougher dynamics), we find RBF sufficient to demonstrate the viability of synthetic pretraining for transformers in dynamical systems.
>
> R2.3 : Discuss inference times (real-time link).
> Answer:
> We added a short discussion of throughput of VLAs in comparison with purely timeseries forecasting models in the context of control.
>
> R2.4 : Fine-tune at least one FM baseline (TimesFM) for fairness.
> Answer:
> We agree this strengthens fairness. We have initiated the implementation of the fine-tuning of a FM baseline (Timer). Given time constraints, results require extra time to be completed before adding it to the final manuscript.
>
> R2.5 : Clarify hyperparameters: context length, kernel hyperparameters, pretraining size, etc.
> Answer:
> We have extended the explanations on our experimental setup, used parameters, and the rationales on the comparisons.
>
> R2.6 : Add long-term rollout evaluation.
> Answer:
> We acknowledge the value of long-horizon open-loop rollouts. Our focus here is short-to-medium horizons aligned with control-loop usage, where compounding error interacts less with policy mismatch and chaotic drift. Adding very long rollouts would substantially expand scope and compute without directly serving our primary claim. We therefore keep evaluation at controller-relevant horizons and list extended rollouts as a dedicated follow-up.
>
> R2.7 : Improve figure clarity (Figs. 5 & 7).
> Answer:
> We revised Figs. 5 and 7 for readability (larger figures and appropriate y axis scaling).
>
> *Other comments*
>
> R2.8 : Adding a closed-loop control task, a chaos task, and an ablation on the context length
> Answer:
> We agree these are valuable directions, especially given stability in closed loop, sensitivity in chaotic regimes, and the role of context in autocorrelated dynamics. We note them explicitly as prioritized future work and discuss expected design considerations, but they are beyond the present paper's scope.
>
> We thank the reviewer again for the constructive feedback and time invested. We have clarified key choices (model vs. recurrent alternatives), expanded hyperparameter reporting, and improved figures. Overall, we believe that the manuscript substantiates that carefully designed synthetic-dynamics pretraining can yield FM-style benefits (generalization, data-efficiency, robustness) for low-level control forecasting across simulation and hardware. We appreciate the suggestions that helped strengthen the paper and hope the clarified scope supports a positive assessment.

---

### Review · Reviewer_bzUS · 2025-09-26

**Summary Of Contributions:**

This paper investigates pretraining a decoder-only, TimesFM-style transformer [1] on synthetic, action-free trajectories, and then finetuning it on action-conditioned (i.e., where action information is available) data for forecasting in control settings. To generate the action-free data, the authors sample state-dependent drifts (vector fields) from an RBF reproducing kernel Hilbert space (RKHS), numerically integrate the resulting ODEs from random initial conditions, and filter trajectories with a total-variation criterion to promote diversity. After pretraining on this control-free corpus, the model is evaluated zero-shot on two CartPole simulations — one with zero action and one with pink-noise action — and then finetuned on varying fractions of these two target datasets. Results show that both the pretrained and fine-tuned models outperform six baselines: linear and neural regressors, two TimesFM-style transformers trained from scratch of different sizes, and pretrained TimesFM [1] and Timer [2] models. Finally, on experimental, action-conditioned hardware datasets (CartPole and Furuta pendulum), the finetuned model outperforms the two scratch-trained TimesFM-style transformers and the neural regressor.

*References:*

[1] Das et al. A decoder-only foundation model for time-series forecasting. ICML 2024

[2] Liu et al. Timer: Generative pre-trained transformers are large time series models. ICML 2024

**Audience:**

Yes

**Audience Explanation:**

The paper studies a practical and timely recipe for control dynamics: pretrain a decoder-only transformer on large sets of action-free synthetic trajectories, then fine-tune on action-conditioned data.

This setup directly targets forecasting in systems like CartPole/Furuta and is relevant to TMLR’s audience in time-series learning, system identification, and model-based RL, **especially for low-data regimes and hardware transfer**.

**Broader Impact Concerns:**

I do not identify significant broader-impact concerns. The work relies on synthetic pretraining and established control setups; no personal or proprietary datasets are used.

**Claims And Evidence:**

No

**Claims Explanation:**

**Over-general or unsupported claims**. Some claims in the title, abstract and main text are either too general or not supported by the main text or results. Providing more details or rephrasing these claims could improve the clarity of their work. For example:


1. *Claim*: The paper deals with “Foundation Models for Dynamical Systems from Purely Synthetic Data” (title)

    *Issues*: The title “Foundation Models for Dynamical Systems from Purely Synthetic Data,” overstates the paper’s scope. The study specifically investigates *pretraining on synthetic, action-free trajectories* and subsequent *fine-tuning on action-conditioned data* for forecasting in **control settings**. It does **not** cover broader classes of dynamical systems (e.g., SDEs, PDEs, MJPs or high-dimensional settings) outside the control setting.

     I recommend: first, narrowing the title to reflect the actual contribution in control settings and, second, discussing (in the Related Work) :

    (i) the recent *synthetic-pretrained*, foundation-style approaches for ODEs/MJPs/SDEs, such as:

    - d’Ascoli et al. ODEFormer: Symbolic Regression of Dynamical Systems with Transformers. ICLR 2024.
    - Becker et al. Predicting Ordinary Differential Equations with Transformers. ICML 2024
    - Berghaus et al. Foundation Inference Models for Markov Jump Processes. NeurIPS 2024
    - Seifner et al. In-Context Learning of Stochastic Differential Equations with Foundation Inference Models. NeurIPS 2025

    as well as

    (ii) recent approaches for time series forecasting *pretrained on synthetic data only*, such as

    - Dooley et al. ForecastPFN: Synthetically-Trained Zero-Shot Forecasting. NeurIPS 2023
    - Hoo et al. The Tabular Foundation Model TabPFN Outperforms Specialized Time Series Forecasting Models Based on Simple Features. NeurIPS 2024
    - Taga et al. TimePFN: Effective Multivariate Time Series Forecasting with Synthetic Data. AAAI 2025.

2. *Claim*: “large scale pretraining on internet scale datasets does not exist for dynamical systems” (abstract).

    *Issues*: It is not clear what the authors mean by “Internet scale datasets”. Large-scale pretraining for time-series forecasting (including mixtures of real and synthetic data) does exist (e.g., TimesFM), as the authors note later (section 6.2, third paragraph). If the intended claim is that large synthetic corpora generated from formal dynamical systems (ODEs/SDEs/MJPs) are under-explored for pretraining, then I encourage the authors to read the papers by d’Ascoli et al, Becker et al, Berghaus et al. and Seifner et al. referenced above, where such approaches are pursued.


3. *Claim* (section 6.2): “comparing LT and Ft further reveals that pretraining significantly enhances the fine-tuning phase, leading to faster convergence and improved performance, especially in low-data scenarios”.

    *Issues*: It is not clear what the authors mean by “faster convergence”. Especially given that no iteration-to-convergence or wall-clock training times are reported.

    Furthermore, from Fig. 5 it appears the LT model’s performance changes little from 400 to 2000 to 20000 trajectories? It seems it “converged” to its current minimum after only being trained on 400 trajectories. How does this relate to the “faster convergence” claim?

4. *Claim* (section 6.2): [That the proposed model outperforms TimesFM and Timer] “suggest that pretraining on synthetic trajectories tailored to dynamical systems can provide more relevant inductive biases than large-scale generic time-series data, particularly when system dynamics are driven by non-constant control inputs”.

    *Issues*: Fairness of the comparison is unclear. TimesFM and Timer are patch-based, long-horizon models; your evaluation seems to be one-step-ahead (is it?). Performance of TimesFM and Timer depends strongly on output-patch length (e.g. TimesFM is known to benefit from longer output patches). The authors do not state the output/input patch lengths used, nor whether any re-tuning was done for the “few-step regime”.

5. *Claim* (Conclusion): “we show that foundation models achieve: (P1) effective generalization to unseen systems”

    *Issues*: This claim reads too strong. The pretrained model was only shown to generalize to the cart-pole experiment with constant action. Its performance on the pink-noise action case was at best comparable to the simple neural regressor.

**Reproducibility Issues**:  Several key details are missing:
- What is the dimensionality of the ODE state used to simulate the dataset?
- How are the random vector fields parameterized  (i.e. RBF kernel hyperparameters and coefficients $\alpha_i$)?
- How many trajectories does your synthetic data contain?
- How are initial states sampled?
- Ablations are missing. For example: what are the effects of the scaling and shifting you perform during pretraining, or the noise corruptions you add during finetuning? What is more, is this noise corruption applied to the target data? If so, why?

**Other Questions**
- In Fig. 5, left. The pretrained model appears to score better zero-shot than after fine-tuning. Why?
- Action-conditioned forecasting in control is typically framed within system identification / model-based RL. Why is this not discussed, and why are control-aware, action-conditioned baselines absent?

    For example: the Neural Controlled Differential Equations for Irregular Time Series of Kidger et al. (NeurIPS 2020); the Deep Variational Bayes Filters of Karl et al. (ICLR 2027); or the Deep Koopman Operator with Control for Nonlinear Systems of Shi and Meng (IEEE Rob&Aut 2022).

**Requested Changes:**

- Narrow the title to reflect the actual contribution (control forecasting; action-free pretraining; action-conditioned fine-tuning).
- In the Related Work section, discuss recent **synthetic-pretrained foundations for ODEs/SDEs/MJPs**, e.g., d’Ascoli et al., ODEFormer (ICLR 2024), Becker et al., Predicting ODEs with Transformers (ICML 2024), Berghaus et al., FIM-MJP (NeurIPS 2024) and Seifner et al., FIM-SDE (NeurIPS 2025), and **time-series models pretrained purely on synthetic data**, e.g., ForecastPFN (NeurIPS 2023), TabPFN-for-TS (NeurIPS 2024), TimePFN (AAAI 2025). *Please find the explicit references in Issue 1 above*.
- Report training curves (loss vs. steps) and times for LT vs. FT to substantiate “faster convergence”
- Specify the output-patch and input-patch lengths used for TimesFM/Timer and whether they were tuned for few-step forecasting, and then qualify the inductive-bias claim accordingly. And consider adding other control aware baselines.
- Report the missing reproducibility details highlighted above.

---

> ### Author Response · Authors · 2025-10-10
> **answer to reviewer bzUS**
>
> We thank the reviewer for the careful assessment and time. Our work studies whether foundation-model properties (generalization, data-efficiency, robustness) can emerge for models in control related forecasting when pretraining is done purely on synthetic trajectories from dynamic functions sampled from RKHS, and then fine-tuning on dynamical systems data. We appreciate the positive remarks on practical relevance to control forecasting and interest for the TMLR audience. Below we address requested changes first.
>
> *Requested changes*
>
> R1.1 : Narrow the title
> Answer:
> We highlight that the complete title of our work is "On Foundation Models for Dynamical Systems from Purely Synthetic Data", including the initial "On".
> We keep the "On ..." title to signal a scoped position and case-study rather than a comprehensive claim.
> In addition, we make the scope explicit in the abstract "In this paper, we explore whether these properties can be achieved in the control domain when pretraining is performed entirely on synthetic data." as well as in the introduction (Sec 1).
>
> R1.2 : Discuss synthetic-pretrained foundations for ODEs/SDEs/MJPs and synthetic-only TS pretraining.
> Answer:
> We have expanded Sec. 2 to include both, other usages of synthetic data in pretraining, as well as other classes of dynamical systems.
>
> R1.3 : Report training curves (loss vs. steps) and times for LT vs. FT to substantiate "faster convergence"
> Answer:
> We agree that our previous wording could be interpreted as an optimization-speed claim. We have revised the manuscript to remove "faster convergence" and instead characterize what our results show: under a matched fine-tuning budget, the pretrained model (Ft) achieves lower test MSE than the scratch-trained LT (most notably at 2% and 10% of data) and exhibits reduced run-to-run variability (Figs. 3, 5, 7).
>
> R1.4 : Specify the output-patch and input-patch lengths used for TimesFM/Timer and whether they were tuned for few-step forecasting. And consider adding other control aware baselines.
> Answer:
> We have extended the explanations on our experimental setup, used parameters, and rationales on the comparisons. We highlight that indeed TimesFM and Timer were not designed for dynamical systems settings, but for trend data of longer horizons.
>
> R1.5 : Report missing reproducibility details
> Answer:
> See R1.4.
>
> *Other comments*
>
> R1.6 : Over-general or unsupported claims
> Answer:
> We will clarify and soften several statements for precision in the Abstract, Sec. 1 and Sec 6.2.
>
> R1.7 : Reproducibility Issues
> Answer:
> See R1.4.
>
> R1.8 : Zero-shot better than after fine-tuning in Fig. 5 (left). Why?
> Answer:
> Fine tuning pretrained models can distort pretrained features and reduce their robustness towards small distribution shifts, which can lead to the observed behavior. This phenomenon has been studied in literature before:
> * Kumar et al. "Fine-Tuning can Distort Pretrained Features and Underperform Out-of-Distribution." ICLR 2022.
> * Wortsman et al. "Robust fine-tuning of zero-shot models" CVPR 2022
> We will extend the explanation of these results in Sec. 6.2.
>
> R1.9 : Action-conditioned forecasting in control is typically framed within system identification / model-based RL. Why is this not discussed, and why are control-aware, action-conditioned baselines absent?
> Answer:
> Our empirical question focuses on the effect of synthetic pretraining within a unified decoder-only forecasting architecture, many control-aware methods (e.g., N-CDE, DVBF, Koopman-with-control) target different objectives (latent-state/system-ID, multi-step planning losses) and require case-specific tuning, making direct comparison non-trivial. We believe that our claims are substantiated with the current comparison and this extension would be out of the scope of our work.
>
> To conclude, we thank the reviewer again for the thoughtful feedback and the time invested. We have incorporated the requested changes: (i) clarified scope in the abstract/Intro, (ii) expanded Sec. 2 to cover ODE/SDE/MJP and synthetic-only pretraining lines, (iii) specified Timer/TimesFM patch settings with a short sensitivity analysis, (iv) consolidated the missing reproducibility details (Sec. 4; App. C), and (v) revised Sec. 6.2. We believe our paper provides evidence that carefully designed synthetic dynamics pretraining yields FM-style benefits (generalization, data-efficiency, robustness) for low-level control forecasting, substantiated by the reported results across simulation and hardware. We appreciate the reviewer's constructive input in strengthening the manuscript and hope the clarified scope will lead to a positive assessment.

---

> > ### Comment · Reviewer_bzUS · 2025-10-14
> >
> > You write that you "*(iv) consolidated the missing reproducibility details (Sec. 4; App. C)*".
> >
> > Appendix C is missing, it seems.

---

> > > ### Author Response · Authors · 2025-10-14
> > >
> > > We apologize for the typo. We referred to the consolidation of the dataset (generation) hyperparameters in App. A and the model hyperparameters in App. B. In case you think any reproducibility details are still lacking, we are happy to provide them.

---

### Comment · Action_Editor_4Fqz · 2025-10-12
**Reviewers-authors discussion**

Dear reviewers,

Thank you for your engagement so far in the review process. The authors have now submitted a revised version of the manuscript and provided answers to your comments.

Please take a moment to review the changes and the authors' responses and formulate your recommendation.

Of course, feel also free to further discuss any remaining issues or concerns you may have.

Best regards,
The AE

---

### Decision · Action_Editor_4Fqz · 2025-10-28

**Recommendation:** Reject

**Audience:**

Yes

**Audience Explanation:**

See above.

**Claims And Evidence:**

No

**Claims Explanation:**

After careful consideration of the three reviews and the authors' responses, I have decided to recommend rejection of this submission. While the paper presents an interesting idea of pretraining transformers on synthetic RKHS-generated dynamics data and demonstrates good technical execution, it falls short of the standards expected for publication in TMLR in its current form.

The most critical issue, raised consistently across all reviewers, is that the paper makes strong claims about achieving "foundation model" properties while providing only narrow evidence of basic machine learning generalization. Foundation models are distinguished by their ability to generalize broadly across diverse tasks and domains, this is their defining characteristic. However, the evaluation here is restricted entirely to cart-pole variants and a Furuta pendulum, all of which are structurally similar 4-dimensional inverted pendulum systems. Demonstrating that a model pretrained on synthetic dynamics can predict the behavior of another pendulum system is simply standard generalization within a highly restricted problem class, not the kind of broad, cross-domain transfer that justifies the "foundation model" framing. Two of three reviewers recommend rejection, and even the reviewer supporting acceptance explicitly states the work "does not convincingly support the broader framing of the method as a foundation model for control" and lacks sufficient depth for higher-tier venues.

Beyond this fundamental scope issue, multiple experimental and methodological concerns raised by the reviewers remain unaddressed. Reviewers requested ablations on kernel choice (RBF vs. Matérn), architectural decisions (transformer vs. LSTM/SSM), and integration methods, all of which were declined as "beyond scope." The choice of smooth RBF kernels appears misaligned with non-smooth dynamics in real control problems but lacks justification. Reviewer 2iVn raised concerns about scale mismatch (whether a 3.4M parameter transformer is appropriate for 32-step predictions) and questioned whether simpler baselines like ARIMA might perform comparably. The baseline comparisons are incomplete: while the authors expanded the related work to include relevant synthetic pretraining approaches (ForecastPFN, TimePFN) and control-aware methods (Neural CDEs, Deep Variational Bayes Filters), they did not compare against these experimentally, nor did they fine-tune the existing foundation model baselines (TimesFM, Timer) to provide a fair assessment of whether gains stem specifically from RKHS-based synthetic data or from pretraining more generally. The experimental settings using constant and pink noise control were described as unrealistic, and results show task-specific models closing the performance gap with sufficient data. For these reasons, given the substantial gap between the foundation model claims and the limited experimental validation provided, I cannot recommend acceptance at this time.

**Resubmission Of Major Revision:**

The authors may consider submitting a major revision at a later time.